# Sleep Deprivation: Effects on Weight Loss and Weight Loss Maintenance

**DOI:** 10.3390/nu14081549

**Published:** 2022-04-08

**Authors:** Evangelia Papatriantafyllou, Dimitris Efthymiou, Evangelos Zoumbaneas, Codruta Alina Popescu, Emilia Vassilopoulou

**Affiliations:** 1Department of Nutritional Sciences and Dietetics, International Hellenic University, 57400 Thessaloniki, Greece; e.papatriantafyllou@hotmail.com (E.P.); vassilopoulouemilia@gmail.com (E.V.); 2Department of Psychiatry, Division of Neurosciences, School of Medicine, Aristotle University of Thessaloniki, 54124 Thessaloniki, Greece; dimitrisefthy@gmail.com; 3Center of Education and Training in Eating Disorders, 14231 Athens, Greece; diatrofioe@gmail.com; 4Department of Abilities Human Sciences, Iuliu Hatieganu University of Medicine and Pharmacy, 40012 Cluj-Napoca, Romania

**Keywords:** sleep deprivation, weight gain, weight management, dietary intake, metabolism

## Abstract

This narrative review presents the findings from intervention studies on the effects of sleep deprivation on eating habits, metabolic rate, and the hormones regulating metabolism, and discusses their relevance to weight loss efforts. Disturbed sleeping patterns lead to increased energy intake, partly from excessive snacking, mainly on foods high in fat and carbohydrates. The studies focused mainly on the effects of sleep duration, but also of sleep quality, on dietary intake during weight loss trials, and on weight loss maintenance. It is important to explore sleep routines that could enhance the efforts of obese and overweight people to lose weight, maintain their weight loss, and improve their overall health.

## 1. Introduction

Sufficient sleep is essential for maintaining healthy physical, mental, and emotional functioning [1]. Optimal sleep duration is determined by several intra- and inter-individual characteristics. A duration of 7–9 h of sleep a night is considered appropriate to support good health in adults of 18–60 years of age [2,3], with an optimal average of 7.5 h [4,5]. Increasing professional and social demands, the advent of artificial lighting at the turn of the last century, and, more recently, the widespread use of computers and other electronic media [6] have reduced the average duration of sleep from 9 h a night in 1910 to 7.5 h 1975, and less than 7 h today [4]. Sleep deprivation due to “social jet lag” is increasing [7], and it is estimated that one quarter of adults and a larger percentage of children and adolescents are deprived of sleep, as sleeping 5–6 h a night during the week has become increasingly common [4].

Sleep deprivation or sleep loss is multifactorial, and has a variety of consequences [8]. The National Health and Nutrition Examination Survey (NHANES) showed significantly higher rates of obesity in adults who reported an average of less than 7 h a night of sleep [9]. Sleep loss has a negative impact on the process of thinking and on the learning, memory, and recall capacity, and thus on the ability to work efficiently and socialize freely, and results in a general feeling of being “disconnected” from the world [10]. Sleep deprivation is associated with an increased risk of obesity, a poor lipid–lipoprotein profile, type 2 diabetes mellitus (DM), hypertension and other cardiovascular diseases (CVD) [11], and even premature death [12]. Disturbance of the sleep pattern is often associated with long-term unhealthy “Western” dietary habits [12].

A chronic pattern of sleep duration of ≤6 h a night has been associated with a higher body mass index (BMI) [13,14]. Epidemiological and laboratory studies have consistently demonstrated that short sleep duration is a significant risk factor for weight gain and obesity, especially in African-Americans and men [15], contributing to poor health outcomes [16,17]. Restricting sleep for up to 5 days can lead to short-term weight gain [18]. Several cross-sectional studies have indicated that short sleep duration is associated with obesity and the risk of future weight gain in both adults and children [19]. There is evidence that eating and sleeping at unconventional times is associated with a higher risk of obesity and an unfavorable metabolic profile. A higher prevalence of obesity and cardiometabolic dysregulation has been reported in people working on night shifts [20,21,22,23], and in those with changes in the time of sleep between working “days on” and “days off”, work patterns which desynchronize the circadian clock [24,25].

### 1.1. Eating Habits, Obesity, and Sleep Duration

Circadian rhythmicity affects the weight loss process, and has therefore been suggested as a predictor of weight loss effectiveness [26]. Changes in meal timing and sleep disorders both increase the risk of obesity by affecting the dietary content, in both energy and quality, and other lifestyle factors [27,28]. Short sleep duration is reported to be associated with higher energy intake, mainly due to increased consumption of saturated fat, resulting in weight gain and an increase in BMI [29]. It is associated with poor eating habits, including an increase in meals, snacks, and night-time eating, with the consumption of high energy foods, lower intake of fruits and vegetables, and a higher intake of fast foods, sugar, and fats, resulting in an overall higher energy intake and increased BMI [30,31,32,33,34,35]. St-Onge and colleagues [19] suggested that diet can influence night-time sleep propensity, depth, and architecture. They reported that a higher intake of saturated fat and a lower intake of fiber were associated with a lighter, less profound sleep profile, and that increased intake of both sugar and non-sugar carbohydrates was associated with more frequent nocturnal arousal during sleep [19]. Improvement in dietary quality may mitigate the disease risk associated with obesity and impaired sleep quality [12].

### 1.2. Bilateral Associations of Sleep Duration and Dietary Changes: Hormones and Weight Gain

Sleep–wake cycles are strictly controlled by circadian rhythmicity, and exert a strong effect on the circulating levels of ghrelin and leptin, hormones that regulate appetite and caloric intake [36]. Short sleep duration may be associated with an increase in the orogenic hormone ghrelin, which stimulates hunger, and a decrease in the saturating hormone leptin [4,37], leading to increased food intake to combat fatigue or stress, among other possible mechanisms [29,37,38,39,40]. Poor sleep undermines dietary efforts to reduce weight by altering the levels of the appetite-regulating hormones, leading to reduction in dietary compliance [41]. An increase in sleep duration and correction of sleep disorders may be accompanied by a better balance of the hormones that regulate appetite, with enhanced glucose tolerance, and a reduction in the level of cortisol [11].

Grandner and colleagues [5] showed that total sleep time was negatively correlated with fat intake in women [5]. Severe energy restriction is known to interfere with sleep; specifically, Karklin and colleagues [42] reported an increase in the latent time at the start of sleep and a reduction in the time spent in slow-wave sleep (SWS) in 9 overweight women, after four weeks on an 800 kcal diet. In another study, two days of a high-carbohydrate, low-fat diet resulted in shorter SWS and longer rapid eye movement (REM) sleep in 8 normal-weight men, compared with a 2-day low-carbohydrate, high-fat, balanced diet [19].

Calvin and colleagues [29] proposed that people undergoing sleep deprivation tend to gain weight due to increased caloric intake, and that therefore sleep deprivation may provide a strong impetus for the development of obesity. An average increase in energy intake of 200 to 500 kcal/day has been documented after imposed sleep deprivation, compared with normal sleep duration, suggesting that increased sleep duration may be largely due to the weight gain observed after sleep deprivation [18].

Some studies suggest that sleep plays a role in weight loss, and specifically fat loss, during calorie restriction. In a study of people undergoing 14 days of calorie restriction, the participants lost less fat when they spent 5.5 h a day in bed than when they spent 8.5 h a day in bed [40]. In line with this finding, an intervention that included a 4-day controlled diet combined with short sleep (4 h a day in bed) resulted in less weight loss than with normal sleep (9 h a day in bed) [13].

Based on the growing interest in the role of sleep in weight management [43], the aim of this narrative review was to investigate the effects of sleep deprivation in people aiming at weight loss or weight control.

## 2. Materials and Methods

### 2.1. Literature Selection and Eligibility Criteria

A literature search was conducted in the electronic databases of PubMed, Scopus, Google Scholar, Academia, and Web of Science, using the terms: sleep deprivation OR sleep duration OR sleep quality AND weight management OR weight loss OR weight control AND obesity OR overweight. The review focused on clinical trials and randomized controlled trials (RCTs) published in peer-reviewed publications, involving healthy individuals without any chronic non-communicable disease, who were aiming at weight loss or weight maintenance. Relevant outcomes from observational studies, books, metanalyses, and reviews facilitated the interpretation of the results derived from the clinical trials. No date restrictions were set. A first literature search was conducted in December 2021, and the search was up to date as of 18 March 2022.

### 2.2. Study Selection

The flow chart of the selection of clinical trials and RCTs is shown in Figure 1. The online service Rayyan was used for blinding the reviewers during the selection of the studies included in the review. Duplicate records were excluded, titles and abstracts were screened for study eligibility, and full-text articles were reviewed by E.P. and E.V. Articles that were the subject of disagreement were discussed with a third person, D.E. Two additional reviewers, A.C.P. and E.Z., examined the studies in the case of further disagreement, and they were discussed until consensus was reached.

## 3. Results and Discussion

In total, 1539 publications were identified in the literature search. After removing 485 duplicates, 1054 studies were screened and assessed for eligibility, of which 502 were excluded, and a further 68 were dismissed after reading the abstract. Of the remaining 484 studies, only 10 fulfilled the review criteria and were clinical trials or RCTs investigating the association of sleep duration with weight changes or weight control. The 10 studies are summarized in Table 1. Observational trials, systematic reviews, and metanalysis were included and discussed in relation to the outcomes of the clinical trials or RCTs.

### 3.1. Sleep Duration and Weight Loss

Βoth better sleep quality and longer sleep duration were documented to be associated with higher success in weight loss efforts [5,40,41,43,44,45,46,47,48,49,51].

The study of Thomson and colleagues [43] was one of the first to report a relationship between sleep and weight loss success in a large sample of overweight or obese women who participated in an intervention study on weight loss. They conducted an identical 24-month weight-loss trial in two groups of women, who differed only on their sleep duration: <7 h or >7 h of sleep. Both groups followed a multifaceted weight-loss program including a reduced energy diet prescription, recommendations to increase physical activity, and behavioral counseling including sleep modifications. A third control group received general weight-loss counseling from a dietetics professional alone [43]. The study findings suggested that better sleep quantity and quality increased the likelihood of successful weight loss by 33%, in agreement with other, mainly observational, studies [13,14,52].

Conversely, Nam and colleagues [41], in a 6-month lifestyle intervention aiming at weight loss by either diet alone or diet and exercise, concluded that sleep patterns, as these were evaluated with the John Hopkins Sleep Survey [53], improve in obese and overweight people when they lose weight. Body fat loss was suggested as a potential mediator of the beneficial effect of lifestyle interventions on sleep disorders.

Steinberg and colleagues [11] observed improved sleep outcomes, specifically in sleep disturbances and sleep continuity, when weight gain was prevented in obese subjects. Their computer-based study utilized the Interactive Obesity Treatment Approach, which prescribes a tailored behavior change plan, such as increased fruit consumption or physical activity levels. Adequate sleep, determined as 7–8 h daily, was among the goals to be achieved by the study participants. The outcomes of the study were promising, but as the dietary content was not analyzed, a window remains open for further research in this area [11].

There appears to be a reciprocal relationship between sleep duration and weight loss, and restricted sleep appears to impede fat loss. Wang and colleagues [47] performed a randomized trial of 8-week caloric restriction, with or without sleep restriction. A reduction in sleep by one hour or more per week resulted in a lower rate of fat loss in people who were following a hypocaloric diet. Sleep restriction increases hunger and appetite by altering metabolic and endocrine function; glucose and insulin sensitivity decreases and the evening levels of cortisol and ghrelin increase, while leptin decreases [54]. Inadequate sleep is associated with alterations in the neuroendocrine appetite control mechanism, characterized by a reduction in leptin levels and an increase in ghrelin levels, the hormones that promote satiety and hunger, respectively [37].

A reduction in sleep duration may be associated with an increase in physiological hunger cues [55]. The study of Spiegel and colleagues [56] showed that sleep restriction in healthy men of normal weight (mean BMI 23.6 ± 2 kg/m^2^) led to 24% higher hunger ratings with a parallel elevation in ghrelin levels, an increase in appetite, and a 33% increase in the consumption of calorie-/carbohydrate-dense foods. A cross-sectional study of 1495 overweight/obese subjects who were attending an outpatient clinic detected a significant difference in sleep reduction, changes in ghrelin levels, and evening preferences in carriers of the circadian locomotor output cycles kaput (CLOCK) 3111T/C single-nucleotide polymorphism (SNP), with a significant effect on weight loss. The researchers hypothetized that this CLOCK gene may affect a broad range of behaviors relevant to food intake and weight loss [50]. On the other hand, several studies support the hypothesis that extended sleep contributes to better weight control [13,56,57]. In an RCT by Tasali and colleagues, 80 overweight participants with habitual sleep less than 6.5 h were assigned to a 2-week sleep extension intervention. The intervention group significantly reduced their daily energy intake by approximately 270 kcal compared to the control group. No significant changes were measured in total energy expenditure [58].

Several studies highlight the decrease in the satiety hormone leptin [6,59,60] and increase in the hunger signaling hormone ghrelin [56,61,62] under conditions of restricted sleep, when caloric intake is carefully controlled and weight is maintained, although the findings are not always consistent.

In contrast to the studies that carefully controlled the energy intake and documented changes in the levels of leptin and ghrelin with sleep restriction [60,61,62,63], others failed to demonstrate consistent differences in leptin and ghrelin levels, despite controlling the energy balance or ensuring that participants fasted during the overnight sleep period [61,62]. In a longitudinal, clinical, behavioral weight loss intervention study in 316 overweight and obese women, weight loss was not shown to be significantly correlated with sleep duration [64].

### 3.2. Sleep Quality and Weight Loss

In a web-based weight loss intervention study conducted by Shade and colleagues [46], in rural women, the effect of sleep quality was evaluated in relation to weight loss. The intervention focused on healthy eating and activity behaviors, with an emphasis on self-initiation. Those who achieved a weight loss of 5% or greater self-reported better sleep and less sleep disturbances, as well as less pain and blood pressure problems [46]. Furthermore, in the MedWeight study, Yannakoulia and colleagues found a significant association of sleep quality and weight maintenance in men, but not in women, who had previously achieved weight loss of at least 10% [64].

Other observational studies on obese subjects showed that poor sleep quality is associated with an activation of the stress system, raised resting energy expenditure, and a shift from fat oxidation towards carbohydrate oxidation, and increased protein-calorie intake was observed after sleep restriction [48]. Sleep appears to be important in maintaining the BMI during periods of reduced calorie intake, and the amount of sleep helps to maintain the body fat mass during periods of reduced energy intake. Lack of adequate sleep can jeopardize the effectiveness of standard dietary interventions for weight loss that are aimed at a relative reduction in metabolic risk [40].

Insufficient sleep impedes the efficacy of dietary weight loss interventions, by reducing the metabolic rate and maintaining the fat-free mass during periods of low energy intake [40]. In a cross-over study assessing the effect of normal or late sleep, and normal or late meals on food intake, lower levels of ghrelin and glucagon-peptide were observed in the late sleep and late meals patterns, while the leptin level was decreased only in the late meal pattern [40]. Finally, sleep apnea and poor sleep performance were shown to be associated with an increased respiratory quotient (RQ), which is an indicator of substrate oxidation, and has been suggested as a predictor of the oxidation of carbohydrates rather than fat in the case of poor sleep performance, a high RQ-predicted fat accumulation over time in the studies of Hursel and colleagues [65] and Nedeltcheva and colleagues [40].

### 3.3. Sleep and Dietary Intake

Both the duration and quality of sleep affect dietary intake [51]. Calvin and colleagues [29] observed in a cross-sectional survey that participants increased their energy intake in the range of 1178 to 2501 kcal/daily and increased their weight by 6.5 to 22.5 kg when submitted to experimental sleep restriction.

The cross-sectional study of Nedeltcheva and colleagues [40] indicated that people indulged in increased snacking with higher carbohydrate snacks (especially between 7 p.m. and 7 a.m.) when they slept less (i.e., 5.5 h) than usual (i.e., 8.5 h), for a period of 3 weeks [32]. These findings support the hypothesis that reduced sleep duration not only provides increased snacking time, but is also associated with higher carbohydrate intake, thus increasing overall energy intake and, subsequently, reducing the rate of weight loss.

According to St-Onge and colleagues [49], alignment of sleep and meals appears to affect food choices, and thus the energy balance. Sleep timing tended to exert a greater influence on food intake parameters than meal timing, although the effects of sleep timing were influenced by meal timing, as reflected by a significant sleep-to-meal interaction [49]. It is suggested that multifactorial mechanisms mediate the association between sleep duration and dietary intake, via changes in the leptin and ghrelin levels and the hedonic pathways, in the case of prolonged modification of food intake hours [66].

## 4. Conclusions

This narrative review highlights current evidence on the effects of poor sleep on weight management. Disturbed sleeping patterns, in terms of both quantity and quality, have been documented to lead to an increase in energy intake, mainly from snacking, especially on foods rich in fat and carbohydrates. The relationship between sleep and weight loss seems bi-directional, and although studies up to date use different intervention protocols and/or outcome measures, there is an evident dysregulation of the neuroendocrine appetite control system during sleep deprivation that alters the metabolic rate, with a negative impact on weight maintenance or weight loss interventions [13,67]. Further research should focus on determining the most effective sleeping patterns for optimal hormonal and metabolic function to facilitate and maintain weight loss in overweight and obese individuals, without overlooking the age-related differentiations due to neuroendocrine and lifestyle changes.

## Figures and Tables

**Figure 1 nutrients-14-01549-f001:**
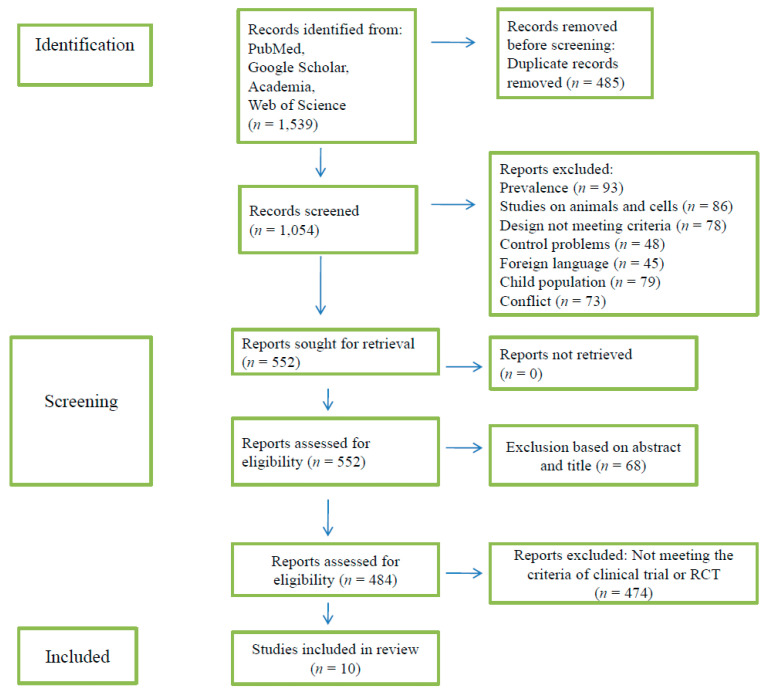
Flow chart for selection of studies on sleep deprivation and weight management.

**Table 1 nutrients-14-01549-t001:** Studies on sleep deprivation and weight management.

Study Title/Doi	Authors	Year	Intervention Description	Duration	Study Design	*N*	Study Sample	Stated Primary Outcome(s)
Insufficient sleep undermines dietary efforts to reduce adiposity/doi:10.7326/0003-4819-153-7-201010050-00006 [32]	Nedeltcheva et al.	2010	Caloric restriction (90% of resting metabolic rate at the time of screening) with 8.5 or 5.5 h of night-time sleep opportunity	2 weeks	RCT	*N* = 10 3 women	Mean age 41 ± 5 years Mean BMI 27.4 kg/m²	Sleep curtailment decreased the proportion of weight lost as fat by 55% (1.4 vs. 0.6 kg with 8.5 vs. 5.5 h of sleep opportunity, respectively; /*p* = 0.043), and increased the loss of fat-free body mass by 60% (1.5 vs. 2.4 kg; *p* = 0.002) Accompanied by markers of enhanced neuroendocrine adaptation to caloric restriction, increased hunger, and a shift towards oxidation of less fat
Lifestyle intervention for sleep disturbances among overweight or obese individuals/doi:10.1080/15402002.2015.1007992 [41]	Nam et al.	2016	Weight loss diet program (600 kcal deficit/day) (D) or diet combined+ with supervised exercise training (D + E) (non-exercise days: 600 kcal deficit/day exercise days: ~energy expenditure from exercise 250 kcal–350 kcal recommended dietary deficit Exercise 3 non-consecutive days/per week	24 weeks	RCT	*N* = 77 60 Women	D: Mean age 56.37 ± 7.17 years Mean BMI 34.11 ± 4.49 kg/m² D+E: Mean age 53.26 ± 8.17years BMI 34.77 ± 5.02	At 6 months: both groups improved from baseline (*p* < 0.05 for all), groups did not differ in changes in body weight (*p* = 0.61), abdominal total fat (*p* = 0.92), and sleep disturbances (*p* = 0.16) Reduction in sleep disturbance score associated with reduction in BMI (*p* < 0.01), abdominal subcutaneous fat (*p* < 0.01), abdominal total fat (*p* < 0.01), and depressive symptoms (*p* < 0.05) Reduction in depressive symptoms associated with improvement in sleep disturbances (*p* < 0.05) and mental composite score on the SF-36 (*p* < 0.05) Adherence to exercise sessions associated with reduction in abdominal subcutaneous fat, BMI, and improved fitness (*p* < 0.05)
Relationship between sleep quality and quantity and weight loss in women participating in a weight-loss intervention trial/doi:10.1038/oby.2012.62 [43]	Thomson et al.	2012	Weight-loss program with energy reduced diet prescription, recommendations to increase physical activity and behavioral counseling	96 weeks	RCT	*N* = 245 women	Women of mean aged 45.5 ± 10.4 years Mean BMI 33.9 ± 3.3 kg/m²	87.4% demonstrated some weight loss (i.e., ≥1 kg) at 6 months 73.1% demonstrated some weight loss at 24 months Better subjective sleep quality increased by 33% the likelihood of weight-loss success (RR 0.67; 95% CI 0.52–0.86)
Behavioral mediators of reduced energy intake in a physical activity, diet, and sleep behavior weight loss intervention in adults/doi:10.1016/j.appet.2021.105273 [44]	Fenton et al.	2021	Move, eat, and sleep: a multiple-behavior-change weight loss intervention Three intervention groups (wait-list control, traditional, enhanced) Physical activity intervention: moderate vigorous intensity physical activity (150 min of moderate or 75 min of vigorous intensive physical activity per week) Dietary intervention: personalized daily energy intake target of 2000 kJ less than their estimated daily energy requirement Sleep intervention: information about the importance of overall sleep health (not just duration); guidance on sleep hygiene, cognitive and behavioral strategies to help in achieving adequate quantity, consistent timing, and improved quality of sleep	24 weeks–48 weeks	RCT	*N* = 116 70% female 81 (70%) completed the six-month assessment	Mean age 44.5 years Mean BMI 31.7 kg/m²	Significant decrease in energy intake, with the pooled intervention group consuming a mean of 1011 less kJ per day than the control group (*p* < 0.05) Significant association/s at six months between total daily EI and minutes per week of physical activity, EI from nutrient-dense foods, energy-dense, nutrient-poor foods, total fat, saturated fat, carbohydrate, protein, and alcohol intake: significant intervention effect on EI at six-months partially mediated by reduced fat intake and reduced consumption of energy dense, nutrient-poor foods
Sleep and health-related factors in overweight and obese rural women in a randomized controlled trial/doi:10.1007/s10865-015-9701-y [45]	Shade et al.	2016	The “Women Weigh-In for Wellness” trial was designed to promote healthy eating, physical activity, and weight loss	24 weeks	RCT	*N* = 221 women	Mean age 54.5 ± 7.0 years Mean BMI 34.6 ± 4.2 kg/m² Mean age 40.8 years Mean BMI 38.5 kg/m²	Self-reported association between sleep disturbance, pain interference and other variables Seep disturbance scores associated only with pain interference scores (*p* < 0.05) Pain interference score associated with higher weight (*p* < 0.05) and BMI (*p* < 0.05) and weak to moderately with older age, higher weight, waist circumference, and systolic, but not diastolic blood pressure Weak relationship between longer objectively measured percent sleep duration and weight loss
Influence of sleep restriction on weight loss outcomes associated with caloric restriction/doi:10.1093/sleep/zsy027 [46]	Wang et al.	2018	Caloric restriction (CR) * alone, or combined with sleep restriction (SR) (reduction in sleep by 90 min on 5 nights and sleep ad libitum on the other two nights each week) * Daily calorie intake to 95% of measured resting metabolic rate	8 weeks	RCT	CR: *N* = 15 12 Females CR+SR: *N* = 21 17 females	CR: age 45.0 ± 5.7 years and BMI 31.3 ± 3.3 kg/m² or weight 88.1 ± 8.8 Kg CR+ SR: age 45.3 ± 6.0 years and BMI 35.1 ± 5.1 kg/m² or weight 99.0 ± 10.9 Kg	No significant change in body weight, body composition, or resting metabolic variables (*p* > 0.16 for time × group interactions) Total mass lost as fat was significantly greater (*p* = 0.016) in the CR group Resting RQ reduced only in CR (*p* = 0.033) fasting leptin level reduced only in CR + SR (*p* = 0.029)
Acute changes in sleep duration on eating behaviors and appetite-regulating hormones in overweight/obese adults/doi:10.1080/15402002.2014.940105 [47]	Hart et al.	2015	Two nights of short (5 h) nights of long (9 h) time in bed sleeping	4 days	RCT	*N* = 12 women	Mean age 41.7 ± 10.3 years Mean BMI 31.0 ± 4.2 kg/m	Significant polysomnographic differences between conditions in total sleep time and sleep architecture (*p* < 0.001). %EI from protein at the buffet increased following short sleep No differences in total EI or measured hormones
Sleep and meal timing influence food intake and its hormonal regulation in healthy adults with overweight/obesity/doi:10.1038/s41430-018-0312-x [48]	St Onge et al.	2019	Controlled food intake and sleep program: normal (00.00-08.00 h) or late (03.30–11.30 h) sleep and meals normal (1, 5, 11, and 12.5 h after waking) or late (4.5, 8.5, 14.5, and 16 h after waking)	16 weeks	RCT Inpatient crossover study controlled, 2 × 2	*N* = 5	Mean age 25.1 ± 3.9 years Mean BMI 29.2 ± 2.7 kg/m²	significant sleep plus meal interaction on energy intake (*p* = 0.035) and a trend for fat and sodium intake (*p* < 0.10) Overnight ghrelin levels higher under normal sleep and meal conditions than late (*p* < 0.005) but lower when combined (*p* < 0.001) Overnight leptin levels higher under normal meal conditions (*p* = 0.012). Significant sleep plus meal interaction on ghrelin (*p* = 0.032) and glucagon-like peptide 1 (*p* = 0.041) levels, but not leptin (*p* = 0.83), in response to a test meal
Efficacy of a multi-component m-health diet, physical activity, and sleep intervention on dietary intake in adults with overweight and obesity: a randomized controlled trial/doi:10.3390/nu13072468 [49]	Fenton et al.	2021	Multi-component weight loss intervention targeting diet, physical activity The traditional intervention group targeted change in dietary and physical activity behaviors The enhanced intervention group targeted change in dietary behaviors, physical activity, and sleep health Increase in daily steps, moderate-to-vigorous intensity physical activity, and resistance training Emphasis on the importance of sleep duration and quality, with daily sleep hygiene practices	24–96 weeks	Randomized Controlled Trial randomly allocation (1:1 ratio)	*N* = 116 70% females	Mean age 44.5 years Mean BMI 31.7 kg/m²	At 12 months, the enhanced intervention group reported improved dietary intake relative to the traditional group: the enhanced group reported higher % EI from nutrient-dense foods and protein and lower % EI from fried/take away foods, baked sweet products, and packaged snacks Weight loss intervention reduced total energy and sodium intake’ with increased fruit intake at six months
Effect of sleep extension on objectively assessed energy intake among adults with overweight in real-life settings: a randomized clinical trial/doi:10.1001/jamainternmed.2021.8098 [50]	Tasali et al.	2022	Sleep extension group: extend their bedtime to 8.5 h Control group: baseline-habitual sleep	4 weeks 4 weeks	RCT	*N* = 80 Control group women = 19 Sleep extension group women = 20	Mean age Control group 30.3 age Mean age sleep extension group 29.3 age Mean BMI in both groups 28.1 kg/m²	The intervention group reduced significantly their daily energy intake by approximately 270 kcal compared to the control group; no significant changes were measured in total energy expenditure

RCT: randomized controlled trial, RR: relative risk, CI: confidence interval, BMI: body mass index, RQ: respiratory quotient, EI: energy intake.

## Data Availability

Not applicable.

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
