# Peer review of "Sleep Deprivation: Effects on Weight Loss and Weight Loss Maintenance"

_nutrients, 2022, doi:10.3390/nu14081549_

Round 1

Reviewer 1 Report

Dear authors,

This is a very interesting and important topic, given that poor sleep and obesity are both global health concerns linked to a host of health problems, and are both modifiable risk factors. I have several concerns about the methods used in the review and how evidence is presented, or often not presented, in the review. I also am perplexed that you refer to this as a systematic review thought out the review, as it is not a systematic, but rather, a narrative review.  Your dedication and interest in the research are is clear, but I have many concerns about the review itself. Please see the comments on the attached PDF.

Reviewer 2 Report

This review is a helpful collection of information about the relationship between sleep deprivation and weight loss. The authors do a great job of summarizing the information, but I would like to see more critical evaluation in the discussion. Thank you for your diligence.

Introduction: Line 25 reads that there is no objective way to measure amount of sleep. I believe the authors are trying to say that there is no objective way to measure the ideal amount of sleep an individual needs for optimal health. Neither of these statements are correct; I suggest deleting the statement.

Results: The results could be organized in a more straight-forward fashion. The introduction was very clear with sub-sections. I would recommend organizing the results by sub-sections or at least by topic rather than by study/author within this section.

Discussion/Conclusion: These sections reiterate the same information as the Introduction and Results section. Please think about what these findings mean for weight loss. All the studies included in this review qualified as clinical trials, but the discussion of the relationship between sleep and weight is portrayed passively. How can someone who is attempting to lose weight or improve sleep increase their chance of success through the information provided in this review?

Funding through Conflict of Interest sections have not been completed.

Round 2

Reviewer 1 Report

Dear authors, 

Thank you for your revisions. Unfortunately, there is no response from the authors to my comments or queries (no responses for any of my comments or queries were provided or attached- only to reviewer 2). With that, I have no way of knowing your thoughts or responses to my previous comments. The review is much improved, nevertheless, and I have only minor comments and suggested edits. However, I await responses to my previous comments (still await) and questions, as well as the few I have added here.

Author Response

please see the archive, it has the correct pdf with comments for round 1 and 2 with our apologies for the wrong file submitted last time

Round 3

Reviewer 1 Report

Dear authors, thank you for your thoughtful revisions and responses to my comments and concerns..